# Factors Predicting Myocardial Recovery After Hospitalization for De Novo Heart Failure with Reduced Left Ventricular Ejection Fraction: Results from the COMFE Registry

**DOI:** 10.3390/biomedicines13051143

**Published:** 2025-05-08

**Authors:** Víctor Donoso-Trenado, Óscar Otero-García, Raquel López-Vilella, Pablo de la Fuente López, Julia Martínez-Solé, Carlos Yebra-Pimentel Brea, Borja Guerrero-Cervera, Javier Adarraga Gómez, Sara Huélamo-Montoro, Guillermo Gallego-Latorre, David García-Vega, Inés Gómez-Otero, Luis Martínez-Dolz, Jose Ramón González-Juanatey, Luis Almenar Bonet

**Affiliations:** 1Heart Failure and Transplant Unit, Hospital Universitari i Politècnic La Fe, Av Fernando Abril Martorell, Number 106, 46026 Valencia, Spain; 2Cardiology Department, Hospital Universitari i Politècnic La Fe, 46026 Valencia, Spain; 3Cardiology Department, Hospital Clínico Universitario de Santiago, 15706 Santiago de Compostela, Spain; 4Instituto de Investigación Sanitaria de Santiago de Compostela (IDIS), 15705 Santiago de Compostela, Spain; 5Centro de Investigación Biomédica en Red de Enfermedades Cardiovasculares (CIBERCV), Instituto de Salud Carlos III, 28029 Madrid, Spain

**Keywords:** de novo heart failure with reduced ejection fraction, predictors of recovery, improved ejection fraction, NT-proBNP, morbidity and mortality

## Abstract

**Background/Objectives**: Patients hospitalized for de novo heart failure with reduced ejection fraction (HFrEF) may experience improvement in left ventricular function, a phenomenon associated with improved morbidity and mortality outcomes. However, the factors influencing this improvement remain unclear. This study aimed to investigate the association between clinical and therapeutic factors and short-term improvement or recovery of left ventricular ejection fraction (LVEF) in patients hospitalized with newly diagnosed heart failure with reduced ejection fraction (HFrEF). **Methods**: This was a prospective observational study conducted in two referral centers in Spain. All patients admitted with de novo HFrEF between March 2021 and December 2023 were included. Improved myocardial function (HFimpEF) was defined as an initial LVEF ≤ 40% and a follow-up echocardiogram showing LVEF > 40%, with an increase of ≥10 points from baseline. **Results**: In total, 157 patients (63.3%) met the criteria for HFimpEF. Among the various etiologies of heart failure, significant differences were found between groups for tachycardiomyopathy (HFimpEF: 29.3% vs. non-HFimpEF: 13.1%, *p* = 0.006), valvular (HFimpEF: 7.6% vs. non-HFimpEF: 1.1%, *p* = 0.05), and ischemic (HFimpEF: 17.2% vs. non-HFimpEF: 43.9%, *p* < 0.0001) etiologies. Multivariate analysis showed that non-ischemic etiologies significantly favored myocardial improvement compared to ischemic cardiomyopathy. NT-proBNP values were consistently higher in the non-HFimpEF group at all time points measured with statistically significant differences, except at admission. Event-free survival curves (hospitalization for HF, worsening HF, and all-cause mortality) diverged early, showing statistically significant differences between groups. **Conclusions**: Overall, 63% of patients hospitalized for de novo HFrEF achieved myocardial improvement within an average of 3–4 months, with improvement favored by valvular and tachycardiomyopathy etiologies. This improvement has a significant prognostic impact.

## 1. Introduction

Heart failure (HF) is a highly prevalent syndrome that accounts for a significant number of hospitalizations and poses a considerable burden on morbidity and mortality. Additionally, it leads to a substantial increase in healthcare costs [1,2].

A large proportion of patients hospitalized for heart failure decompensation have no prior HF diagnosis and are classified as having de novo HF. Traditionally, HF is categorized based on left ventricular ejection fraction (LVEF): reduced ejection fraction (HFrEF) when LVEF ≤ 40%, mildly reduced ejection fraction (HFmrEF) when LVEF is 41–49%, and preserved ejection fraction (HFpEF) when LVEF ≥ 50% [1]. The therapeutic approach for HFrEF is multifaceted and aims to alleviate symptoms, reduce serum biomarkers, improve hemodynamic parameters, shorten hospital stays, prevent readmissions, and decrease all-cause mortality. Advances in the treatment and follow-up of these patients have contributed to the improvement and reversibility of dysfunction in some HFrEF cases. This has led to the identification of a new subgroup of patients with improved or recovered LVEF (HFimpEF) [3,4]. Improvement in LVEF is associated with better prognoses [3,4,5,6,7,8,9,10,11]. However, recovery of LVEF is not always possible, and the factors or circumstances that promote it in a first episode of HF remain unclear. Available data are scarce and primarily derived from studies of patients with chronic outpatient HF.

The hypothesis of this prospective study was that certain variables related to the patient’s hospitalization would be associated with LVEF recovery and could be objectively identified in a study encompassing a large number of patients. Identifying these variables would help to better define patient prognosis and the complications associated with HF progression.

Thus, the objective of this study was to analyze whether specific analytical, clinical, and therapeutic parameters are associated with short-term LVEF improvement/recovery in patients hospitalized with de novo HFrEF. Secondary objectives included evaluating the evolution of NT-proBNP levels and analyzing morbidity and mortality during follow-up in these patients.

## 2. Materials and Methods

### 2.1. Study Population

This prospective observational study was conducted in two referral centers in Spain. All consecutive patients admitted with a diagnosis of de novo heart failure with LVEF ≤ 40% between March 2021 and December 2023 were included. Patients with a prior diagnosis of HF, those with an initial LVEF > 40%, pediatric patients, and those who died during hospitalization were excluded. Additionally, patients without a follow-up echocardiographic assessment were excluded. The total number of patients recruited for the study was 370, of which 248 met the inclusion criteria.

This analysis represents an interim evaluation of an open, long-term prospective database. The database includes clinical, echocardiographic, prognostic, therapeutic, and biomarker variables.

The study was approved by the Biomedical Research Ethics Committees of both participating centers, and the principles outlined in the Declaration of Helsinki for medical research involving human subjects were followed.

### 2.2. Definition of Improved LVEF (HFimpEF) [3,4,12]

(1)Initial LVEF ≤ 40%;(2)Follow-up LVEF > 40%;(3)An increase of ≥10 percentage points between baseline and follow-up assessments.

### 2.3. Guideline-Directed Medical Therapy and Etiology-Specific Interventions

All patients received pharmacological treatment in accordance with current clinical practice guidelines for heart failure with reduced ejection fraction, including the four foundational pillars: renin–angiotensin system inhibitors, beta blockers, mineralocorticoid receptor antagonists, and SGLT2 inhibitors. In addition, specific treatments were administered according to the underlying etiology, when clinically indicated. Patients with tachycardiomyopathy secondary to atrial fibrillation were managed with rhythm control strategies, including electrical cardioversion or catheter ablation in selected cases. Patients with valvular heart disease were assessed by the heart team and, when appropriate, underwent surgical or percutaneous valve interventions. For patients with ischemic cardiomyopathy, coronary angiography was performed during hospitalization, and revascularization (either percutaneous or surgical) was carried out according to guideline recommendations and institutional protocols.

### 2.4. Statistical Analysis

Initially, differences between groups with and without improved LVEF were analyzed for statistical significance (*p*-value < 0.05). For categorical variables, the chi-square test was applied. For numerical variables, the Student’s *t*-test was used if the normality assumption was met; otherwise, the Wilcoxon–Mann–Whitney test was applied.

A logistic regression model was employed, where LVEF improvement served as the dependent variable, and other variables were considered independent. A stepwise selection method was used to identify relevant variables for the final model estimation.

The median and interquartile range of NT-proBNP levels at different time points were calculated. The Wilcoxon signed-rank test for paired data was used to assess whether NT-proBNP levels at each time point were higher than at the preceding time point.

Finally, Kaplan–Meier survival curves were constructed to evaluate morbidity and mortality based on cardiac recovery. The log-rank test was used to assess statistical differences between groups.

## 3. Results

### 3.1. Baseline Characteristics

Of the 248 patients included in the study, 157 (63.3%) met the criteria for HFimpEF on follow-up echocardiography, while 91 patients (36.7%) had persistent reduced LVEF. The baseline characteristics of both groups are presented in Table 1. The HFimpEF group was slightly younger (69 vs. 71 years), with a predominance of males in both groups (76% vs. 75%). Regarding comorbidities, the non-HFimpEF group had a higher prevalence of cardiovascular risk factors, although differences were not statistically significant, except for chronic kidney disease (more frequent in the non-recovery group) and peripheral artery disease (trend toward significance).

The average length of hospital stay was similar between the groups (7 days). No significant differences were found in renal function parameters, potassium, or hemoglobin levels at discharge. The average LVEF on admission was also similar between the groups (28%). A significantly higher percentage of revascularization for ischemic heart disease during hospitalization was observed in the non-recovery group (18% vs. 6%). Although not statistically significant, the use of the four foundational pillars of heart failure therapy at discharge was more frequent in the HFimpEF group (66% vs. 58%). Analyzing individual drug classes, renin–angiotensin system inhibitors and mineralocorticoid receptor antagonists were more frequently used in the HFimpEF group, beta blockers were less frequent, and SGLT2 inhibitors showed similar usage between groups.

After discharge, the average time to therapy optimization was similar in both groups (110 vs. 117 days). Statistically significant differences were observed in the glomerular filtration rate (70 mL/min/1.73 m^2^ in HFimpEF vs. 63 mL/min/1.73 m^2^ in non-HFimpEF) and follow-up LVEF (51.6% vs. 32.9%). The use of the four foundational pillars remained more frequent in the HFimpEF group, although differences were not statistically significant (72% vs. 64%). Individually, all drug classes were more frequently used in the HFimpEF group, except for beta blockers.

### 3.2. Heart Failure Etiology

Significant differences were observed in the HF etiology between the two groups:Tachycardiomyopathy: 29.3% in HFimpEF vs. 13.1% in non-HFimpEF (*p* = 0.006).Valvular: 7.6% in HFimpEF vs. 1.1% in non-HFimpEF (*p* = 0.05).Ischemic: 17.2% in HFimpEF vs. 43.9% in non-HFimpEF (*p* < 0.0001).

Independent Prognostic Factors for LVEF Improvement

Multivariate analysis was performed to assess variables predicting ventricular function improvement. Categorical variables, compared against ischemic etiology as a reference, revealed that non-ischemic etiologies (valvular, tachycardiomyopathy, hypertensive, idiopathic, and other causes) significantly predicted myocardial improvement (Figure 1). The use of the four foundational pillars after therapy optimization showed a trend toward significance. Age, male sex, and peripheral artery disease showed trends toward association with non-recovery of LVEF.

Cardiac magnetic resonance imaging (CMR) was performed in 102 patients (41.1%) as part of the etiologic evaluation. Data on late gadolinium enhancement (LGE) were available in 71 of them. Among patients with positive LGE (n = 29), only 14 (48.3%) met criteria for LVEF recovery (HFimpEF), while in those without LGE (n = 41), 32 (78.0%) showed recovery. This difference was statistically significant (*p* = 0.020), suggesting that the absence of myocardial fibrosis detected by LGE may be associated with a greater probability of myocardial recovery.

Regarding valvular etiology, all patients with severe valvular disease included in the study underwent treatment, either surgical or percutaneous. Specifically, eight patients had severe aortic stenosis with left ventricular dysfunction: six were treated with transcatheter aortic valve implantation (TAVI), and two underwent surgical valve replacement. In addition, two patients with severe aortic regurgitation and two with severe mitral regurgitation were treated surgically. The only patient with valvular disease who did not experience LVEF recovery had rheumatic multivalvular disease, with combined severe aortic and mitral valve lesions and tricuspid involvement.

### 3.3. NT-proBNP Evolution During Follow-Up

At all time points (admission, discharge, 1 month, 6 months, 12 months, and 24 months), NT-proBNP levels were consistently higher in the non-HFimpEF group. Statistically significant differences were observed between groups at all time points, except at admission (Figure 2). Over time, NT-proBNP levels progressively decreased in both groups. Paired data analysis revealed no significant differences in NT-proBNP levels between discharge and 1 month, and between 6 and 12 months, in the non-HFimpEF group (Table 2).

### 3.4. Morbidity and Mortality in HFimpEF

The mean follow-up period was 16.4 months. The HFimpEF group had a significantly lower incidence of HF-related hospitalizations (7% vs. 28.6%, *p* < 0.0001; Table 3). Emergency department visits and all-cause mortality were more frequent in the non-HFimpEF group but did not reach statistical significance.

Event-free survival curves (HF hospitalization, decompensation, or all-cause mortality) diverged from the first days of follow-up, showing significant differences between groups (*p* < 0.0001; Figure 3). Event-free survival rates for the HFimpEF group were 95% at 4 months, 86% at 1 year, and 83% at 2 years, compared to 80%, 70%, and 57%, respectively, in the non-HFimpEF group.

## 4. Discussion

Heart failure (HF) remains a highly prevalent syndrome, accounting for a significant number of hospitalizations and contributing substantially to patient morbidity and mortality [1,2]. A large proportion of patients admitted for HF for the first time present with de novo HFrEF. These patients receive pharmacological treatment based on the “four foundational pillars” of HF therapy (along with specific treatments according to etiology), which have been shown to improve outcomes by reducing symptoms, rehospitalizations, and mortality [1,4]. Over time, following treatment, some patients experience improvement in LVEF (HFimpEF), which is associated with better prognoses [3,4,5,6,7,8,9,10,11]. However, the factors that facilitate myocardial recovery are not fully understood.

In this study, a large series of patients hospitalized with de novo HFrEF in two Spanish referral centers was analyzed. We found that 63% of these patients experienced functional improvement within an average of 3–4 months, with etiologies such as tachycardiomyopathy and valvular heart disease favoring recovery. NT-proBNP proved to be a useful biomarker for monitoring these patients. Improvement in myocardial function has significant prognostic implications for this patient group.

### 4.1. Comparison with Previous Studies

The scientific literature to date shows substantial variability in the definition of myocardial recovery. This inconsistency complicates comparative analysis across studies. In 2020, Wilcox et al. proposed a definition for HF with recovered LVEF due to the lack of consensus and the need to distinguish this group from HFrEF or HFpEF. Their proposed definition aligns with most studies in the literature and was adopted in this study [3]. The 2021 European Society of Cardiology (ESC) guidelines for HF suggest that patients with initial LVEF ≤ 40% who later achieve LVEF ≥ 50% should be referred to as HFimpEF rather than HFpEF [1]. Similarly, the 2021 universal definition of HF by American, European, and Japanese societies defines HFimpEF with criteria consistent with this study [12]. The 2022 American guidelines further consolidated this definition and recommended using the term “improved” rather than “recovered” [4].

In this series, 63.3% of patients with de novo HFrEF met the criteria for HFimpEF on follow-up echocardiography. The literature reports varying proportions of LVEF improvement, likely due to differences in definitions, patient populations (outpatients vs. inpatients), and study designs, for instance:Basuray et al. reported a 10% recovery rate in an outpatient cohort, using a definition of LVEF > 50% that had previously been <50% [13].The IMPROVE-HF registry found a 29% improvement rate in outpatient settings, with an absolute LVEF increase of ≥10 points [5].Agra Bermejo et al. reported a 52% recovery rate at 1 year in outpatients [6].Abe et al. observed a 41% recovery rate among hospitalized patients, although their criteria required LVEF >50%, and these were not de novo cases [7].

In our study, the high proportion of HFimpEF can be explained by the inclusion of patients with de novo HF requiring hospitalization. This particular subset is underrepresented in the literature, and our findings provide valuable insights.

### 4.2. Prognostic Factors

Non-ischemic etiologies such as tachycardiomyopathy and valvular disease were strong predictors of LVEF improvement. In contrast, ischemic cardiomyopathy was associated with a lower likelihood of recovery. These findings align with previous studies:Marcusohn et al. reported a 64% recovery rate in patients with atrial fibrillation-related tachycardiomyopathy, with ischemic heart disease and worse baseline LVEF as negative predictors [14].Abe et al. found valvular disease and atrial fibrillation to be associated with recovery in hospitalized HF patients with reduced LVEF [7].

Several studies have consistently shown that ischemic cardiomyopathy is less likely to result in myocardial recovery, even after complete revascularization [3,5,6,8,15,16].

Other predictors described in the literature include younger age, female sex, absence of left bundle branch block, higher baseline blood pressure, and the use of renin–angiotensin–aldosterone system inhibitors and beta blockers [3,6,7,8,17,18]. In our study, some of these factors showed trends but did not reach statistical significance, possibly due to the sample size or the distinct characteristics of de novo HF patients compared to chronic HF populations.

It is important to highlight that all patients, including those with tachycardiomyopathy or valvular disease, were managed with guideline-directed medical therapy unless contraindicated. Specific interventions addressing the underlying etiology were implemented as complementary strategies rather than alternatives to pharmacological treatment. Therefore, the observed improvement in LVEF likely reflects the synergistic effect of both optimal medical therapy and targeted etiologic interventions.

Recent studies have contributed to a deeper understanding of the mechanisms associated with left ventricular reverse remodeling. In particular, Paolisso et al. demonstrated that coronary microvascular dysfunction, assessed invasively, was an independent predictor of reduced myocardial recovery in patients with de novo heart failure and non-obstructive coronary artery disease [19]. Similarly, in a separate study, the same group showed that among diabetic patients with aortic stenosis undergoing transcatheter aortic valve implantation, treatment with SGLT2 inhibitors was associated with enhanced LV recovery and improved clinical outcomes [20]. These findings reinforce the concept that myocardial recovery is influenced by multiple, modifiable clinical factors and underscore the importance of individualized treatment strategies.

### 4.3. Role of NT-proBNP

In HFrEF, greater reductions in natriuretic peptide levels are associated with improvements in LVEF, reductions in ventricular volumes, and better clinical outcomes [21]. In our study, NT-proBNP levels were consistently lower in the HFimpEF group at all time points after discharge, with statistically significant differences. This biomarker demonstrated a progressive decline in both groups over time. Although limited data are available on the role of NT-proBNP in predicting LVEF improvement in de novo HF, our findings highlight its potential value for prognostication.

### 4.4. Morbidity and Mortality

Patients with HFimpEF had significantly lower rates of HF-related hospitalizations during follow-up. Although emergency visits and all-cause mortality were less frequent in the HFimpEF group, these differences did not reach statistical significance, likely due to the sample size. In larger cohorts, such as the 1057-patient study by Lupon et al., HFimpEF patients had significantly lower rates of mortality, HF hospitalizations, and sudden cardiac death compared to HFrEF, HFpEF, and healthy controls [22]. Similar results were reported in the meta-analysis by He et al., which included nine studies and 9491 participants [9].

### 4.5. Limitations

The observational design and sample size of this study may limit the statistical power of the findings. Additionally, variability in the definitions and criteria for HFimpEF across studies hinders direct comparisons. Nevertheless, this study is the first to analyze myocardial improvement in patients with de novo HFrEF. Conducted in two high-volume referral centers, its prospective design ensures robust and reliable data.

## 5. Conclusions

In conclusion, 63% of patients hospitalized with de novo HFrEF experienced significant LVEF improvement within an average of 3–4 months. This improvement was favored by tachycardiomyopathy and valvular etiologies and had a profound prognostic impact, significantly reducing morbidity and mortality. These findings underscore the importance of a personalized approach to managing and monitoring these patients to optimize clinical outcomes.

## Figures and Tables

**Figure 1 biomedicines-13-01143-f001:**
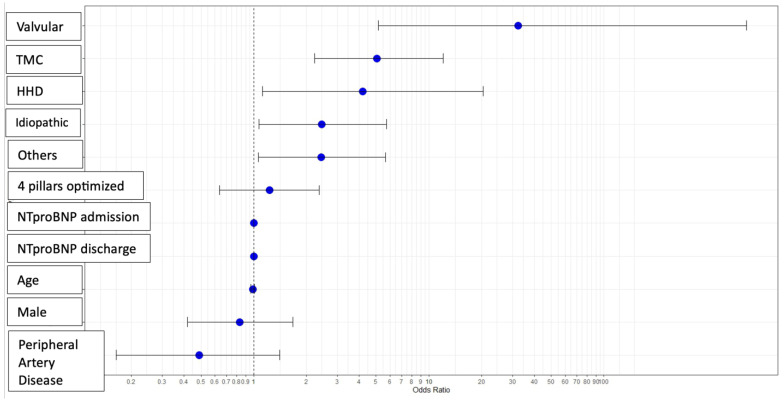
**Predictive variables for the improvement of ventricular function: multivariable analysis.** Abbreviations—HHD: Hypertensive Heart Disease; TCM: Tachycardiomyopathy.

**Figure 2 biomedicines-13-01143-f002:**
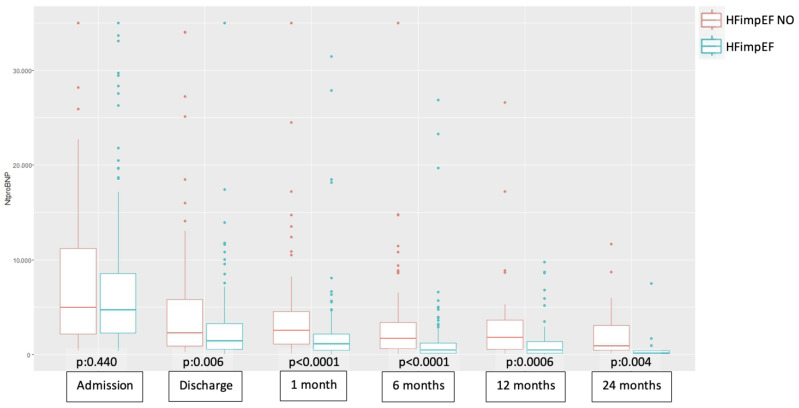
**Boxplot of the comparative evolution of NT-proBNP across study groups at different time points.** Abbreviations—HFimpEF: Heart Failure with Improved Ejection Fraction.

**Figure 3 biomedicines-13-01143-f003:**
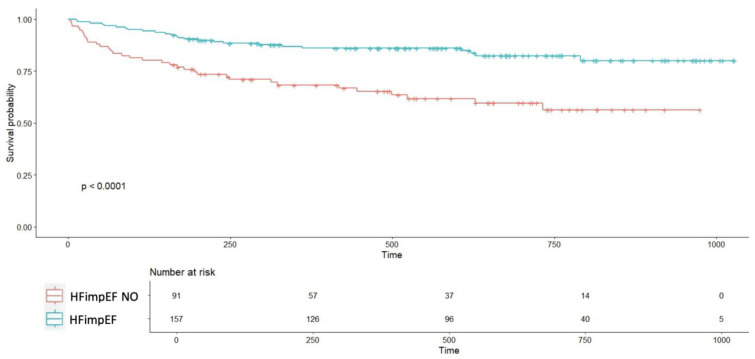
**Event-free survival curve comparing patients with improved ventricular function to those without improvement.** Abbreviations—HFimpEF: Heart Failure with Improved Ejection Fraction. Time (days).

**Table 1 biomedicines-13-01143-t001:** Baseline characteristics of the study population.

Variables	HFimpEF (n = 157)	Non-HFimpEF (n = 91)	*p*-Value
Age (years)	69 (58–76)	71 (61–78)	0.353
Male (n, %)	120 (76.43%)	68 (74.73%)	0.882
BMI (n, %)	26.66 (23.72–30.16)	27.15 (23.55–31.13)	0.527
Hypertension (n, %)	92 (58.60%)	63 (69.23%)	0.126
Dyslipidemia (n, %)	73 (46.50%)	48 (52.74%)	0.414
Diabetes mellitus (n, %)	43 (27.39%)	34 (37.36%)	0.135
Chronic kidney disease (n, %)	28 (17.83%)	29 (31.87%)	0.018
Stroke (n, %)	8 (5.12%)	8 (8.79%)	0.390
COPD (n, %)	11 (7.00%)	10 (10.99%)	0.396
OSA (n, %)	7 (4.49%)	7 (7.69%)	0.444
Peripheral artery disease (n, %)	9 (5.73%)	12 (13.33%)	0.068
Alcohol use (n, %)	44 (28.03%)	21 (23.08%)	0.481
Smoking (n, %)	64 (40.76%)	26 (28.57%)	0.074
Length of hospital stay (days)	7 (5–11)	7 (5–10.5)	0.805
Time to therapy optimization (days)	117 (38.25–186.25)	110 (44.50–172.00)	0.837
Heart failure etiology (n, %)	<0.0001		
- Hypertensive	10 (6.37%)	3 (3.30%)	0.453
- Tachycardiomyopathy	46 (29.30%)	12 (13.19%)	0.006
- Ischemic	27 (17.20%)	40 (43.96%)	<0.0001
- Idiopathic	32 (20.38%)	20 (21.98%)	0.892
- Valvular	12 (7.64%)	1 (1.10%)	0.050
- Other	26 (19.11%)	15 (16.48%)	0.729
4 Pillars at discharge (n, %)	103 (66.03%)	53 (58.24%)	0.277
4 Pillars after optimization (n, %)	110 (72.37%)	57 (64.04%)	0.227
RAAS inhibitors at discharge (n, %)	142 (90.45%)	76 (83.52%)	0.158
Beta blockers at discharge (n, %)	142 (91.03%)	88 (96.70%)	0.150
Mineralocorticoid receptor antagonists at discharge (n, %)	130 (82.80%)	69 (75.82%)	0.244
SGLT2 inhibitors at discharge (n, %)	131 (83.39%)	76 (83.52%)	0.126
RAAS inhibitors after optimization (n, %)	139 (90.26%)	74 (81.32%)	0.070
Beta blockers after optimization (n, %)	147 (93.63%)	88 (97.78%)	0.250
Mineralocorticoid receptor antagonists after optimization (n, %)	128 (82.58%)	74 (81.32%)	0.939
SGLT2 inhibitors after optimization (n, %)	144 (91.72%)	80 (88.89%)	0.611
Creatinine at discharge (mg/dL)	1.10 (0.89–1.37)	1.14 (0.87–1.51)	0.361
eGFR at discharge (mL/min/1.73 m^2^)	65.93 (21.63)	62.00 (24.44)	0.217
Potassium at discharge (mEq/L)	4.2 (3.9–4.50)	4.3 (3.9–4.85)	0.296
Hemoglobin at discharge (g/dL)	14.35 (2.24)	13.99(2.50)	0.260
eGFR after optimization (mL/min/1.73 m^2^)	70.03 (23.33)	63.12 (25.90)	0.041
Potassium after optimization (mEq/L)	4.62 (0.45)	4.55 (0.52)	0.266
LVEF at admission (% mean)	28.90 (6.44)	28.78 (7.38)	0.90
LVEF after optimization (% mean)	51.58 (7.249)	32.916 (6.60)	<2.2 × 10^−16^
ECG rhythm on admission			0.285
- Sinus rhythm (n, %)	104 (60.8%)	67 (39.2%)	
- Atrial fibrillation/Flutter (n, %)	53 (68.8%)	24 (31.2%)	
Bundle branch block on admission			0.004
- No bundle branch block (n, %)	121 (69.9%)	52 (30.1%)	
- Left bundle branch block (n, %)	24 (45.3%)	29 (54.7%)	
- Right bundle branch block (n, %)	10 (55.6%)	8 (44.4%)	
Coronary anatomy (n,%) *			0.268
- Normal coronaries	47 (62.67%)	28 (43.8%)	
- Non-obstructive CAD	18 (24%)	19 (29.7%)	
- Obstructive CAD	10 (13.33%)	17 (26.6%)	
Revascularization during hospitalization (n, %)	10 (6.37%)	17(18.68%)	0.005
Complete revascularization (n/%)	4 (2.55%)	14 (15.38%)	0.001

* Coronary angiography was performed in 75 patients in the HFimpEF group and in 64 patients in the non-HFimpEF group. Abbreviations—BMI: Body Mass Index; CAD: Coronary Artery Disease; COPD: Chronic Obstructive Pulmonary Disease; eGFR: Estimated Glomerular Filtration Rate; HFimpEF: Heart Failure with Improved Ejection Fraction; LVEF: Left Ventricular Ejection Fraction; OSA: Obstructive Sleep Apnea; RAAS: Renin–Angiotensin–Aldosterone System; SGLT2: Sodium-Glucose Cotransporter-2.

**Table 2 biomedicines-13-01143-t002:** Evolution of NT-proBNP over time in patients with and without HFimpEF using the Wilcoxon signed-rank test for paired data.

Group	Comparison	*p*-Value
HFimpEF	Admission > Discharge	<2.2 × 10⁻^16^
Non-HFimpEF	Admission > Discharge	1.188 × 10⁻^11^
HFimpEF	Discharge > 1 Month	0.001
Non-HFimpEF	Discharge > 1 Month	0.774
HFimpEF	1 Month > 6 Months	3.462 × 10⁻⁸
Non-HFimpEF	1 Month > 6 Months	7.932 × 10⁻⁶
HFimpEF	6 Months > 12 Months	0.019
Non-HFimpEF	6 Months > 12 Months	0.133
HFimpEF	12 Months > 24 Months	0.032
Non-HFimpEF	12 Months > 24 Months	0.058

Abbreviations—HF: Heart Failure; HFimpEF: Heart Failure with Improved Ejection Fraction.

**Table 3 biomedicines-13-01143-t003:** Follow-up events by HFimpEF status.

Variable	HFimpEF (n = 157)	Non-HFimpEF (n = 91)	*p*
HF-related hospitalizations (n, %)	11 (7.01%)	26 (28.57 %)	1.037x10^−5^
Emergency department visits for HF (n, %)	6(3.82%)	6 (6.59%)	0.507
All-cause mortality (n, %)	12 (7.64%)	9 (9.89%)	0.707

Abbreviations—HF: Heart Failure; HFimpEF: Heart Failure with Improved Ejection Fraction.

## Data Availability

The dataset is available on request from the authors.

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
