# Peer review of "Factors Predicting Myocardial Recovery After Hospitalization for De Novo Heart Failure with Reduced Left Ventricular Ejection Fraction: Results from the COMFE Registry"

_biomedicines, 2025, doi:10.3390/biomedicines13051143_

Round 1

Reviewer 1 Report

Comments and Suggestions for Authors

The work of Donoso-Trenado et al.  has analyzed 1) if predefined parameters are associated with short-term LVEF improvement/recovery in patients hospitalized with de novo HFrEF, and 2) if NT-proBNP levels can be correlated with morbidity and mortality. 

Minor comments:

  1. The Graphical abstract is a bit confusing regardin to what does the "10 points" within the 3.rd bullet point of the HFimpEF refers to. I would suggest to write "10 percentage points" as the authors already did on line 96 of the manuscript.
  2. The patients have been recruited in two centers over a period of 33 months (March 2021- December 2023). How would the authors comment on the low recruitment rate? Is it linked to the patients adherence to the recruitment itself or more to the follow-up period? 
  3. Were STEMI/NSTEMI patients included in the ischemic heart disease group? If yes, did the authors notice any difference between these two groups?

Major comments:

    1. The authors mention that the patients received the "pharmacological treatment based on the "four foundational pillars" of HF therapy (along with specific treatments according to etiology)". Could the authors mention in the Methods section which was the "specific treatment"?
    2. Could the authors comment on the left ventricular ejection fraction (LV-EF) among patients with tachycardiomyopathy and valvular disease? Did all these patients receive "four foundational pillars" of the heart failure therapy additional to the specific therapy? or did the authors notice an improvement of LV-EF with the "specific therapy" only?

Reviewer 2 Report

Comments and Suggestions for Authors

Donoso-Trenado et al. investigated clinical and therapeutic factors associated with short-term improvement or recovery of left ventricular ejection fraction (LVEF) in patients hospitalized with newly diagnosed heart failure with reduced ejection fraction (HFrEF) in two Spanish referral Centers. Of interest, 63% of patients hospitalized for de novo HFrEF achieved myocardial improvement within an average of 3-4 months, with improvement favored by valvular and tachycardiomyopathy etiologies. NT-proBNP proved to be a useful biomarker for monitoring these patients. Improvement in myocardial function has significant prognostic implications for this patient group.

The study is interesting and timely, with well-optimized pharmacological therapy in line with current ESC guidelines; nevertheless, I have some concerns that need to be addressed before the study could be re-submitted.

  1. The title seems more like the title of a Review. Please remove the question mark and consider changing it to “Factors predicting myocardial recovery after hospitalization for de novo heart failure with reduced left ventricular ejection fraction: results from the COMFE Registry.
  2. Study objective: I would suggest rephrasing as follows “This study aimed to investigate the association between clinical, and therapeutic factors and short-term improvement or recovery of left ventricular ejection fraction (LVEF) in patients hospitalized with newly diagnosed heart failure with reduced ejection fraction (HFrEF)”.
  3. Study population: why did the authors exclude patients who “died during hospitalization”: This would have introduced a bias in the survival analysis.
  4. No sample size calculation is provided.
  5. Study population: what about Tako-Tsubo patients? Were these patients included in the study population?
  6. Table 1, Heart failure etiology (n, %): please consider indicating only one p-value for all, given that the different etiologies correspond to the entire population of the group.
  7. Table 1. I would suggest adding ECG parameters (presence of AF, LBBB/RBBB at admission); I guess the absence of AF/LBBB might be associated with LV recovery or HFimpEF.
  8. Table 1: please report the coronary artery anatomy; how many patients had normal coronary artery/non-obstructive and obstructive coronary artery disease?
  9. In addition to the number of (complete) revascularization during the hospitalization, please provide also the number of patients who performed valvular heart surgery/interventions (AVR/MVR, TAVI, MitraClip).
  10. Can the authors provide more details of echocardiographic parameters (EDVi, LAVi, SysPAP, VHD) – maybe the authors can provide a new Table 2 with LVEF and the other echo parameters?
  11. Any data about global longitudinal strain?
  12. How many patients performed CMR to further investigate HF etiologies? Any data regarding tissue characterization?
  13. Please consider including in the discussion Section a comparison with similar data regarding factors associated with LV reverse remodeling/recovery (PMID: 38108151, 39574095).
Comments on the Quality of English Language

 The English could be improved to more clearly express the research.

Round 2

Reviewer 2 Report

Comments and Suggestions for Authors

The authors have made a substantial effort to respond to the reviewers' comments, and the manuscript has improved.

I have no further comments.